# The Association between the Sense of Coherence and the Self-Reported Adherence to Guidelines during the First Months of the COVID-19 Pandemic in Israel

**DOI:** 10.3390/ijerph19138041

**Published:** 2022-06-30

**Authors:** Anne Marie Novak, Adi Katz, Michal Bitan, Shahar Lev-Ari

**Affiliations:** Department of Health Promotion, School of Public Health, Faculty of Medicine, Tel Aviv University, Tel Aviv 6997801, Israel; annemarie@mail.tau.ac.il (A.M.N.); di.diamond85@gmail.com (A.K.); bennoac@mail.tau.ac.il (M.B.)

**Keywords:** social distancing, sense of coherence, COVID-19, perceived danger, perceived protection, self-reported adherence, COVID-19 guidelines, pandemic behavior

## Abstract

(1) Background: Social distancing became a central strategy employed to limit the spread of the SARS-CoV-2 virus. We explore self-reported adherence (SRA) and factors associated with SRA among Israeli adults at the end of the first national lockdown in Israel. (2) Methods: We conducted a cross-sectional consumer panel survey of 820 Israeli adults aged 18 to 70 in May and June 2020. We collected data on the SRA to the social distancing measures, sociodemographic variables, perceptions of pandemic-related danger and of protection provided by the social distancing measures, as well as Sense of Coherence (SoC). (3) Results: 60% of respondents reported complying with 7 measures. Higher SoC was associated with higher SRA (*p* = 0.04), and was related to income, marital status, age, profession, and education. The SRA was higher among Jews than Arabs (Jews: Mean = 10.5, SD = 4.5; Arabs: Mean = 9.1, SD = 4.1, *p* < 0.001) and among males (Males: Mean = 10.8, SD = 4.7; Females: Mean = 9, SD = 4.1; *p* = 0.003). SoC, perception of protection and perception of danger were associated with higher SRA (*p* = 0.42, *p* < 0.001 and *p* = 0.005 respectively). Single people reported higher levels of SRA than people in relationships (Partnered: Mean = 9.7, SD = 4.2, Non-partnered: Mean = 10.9, SD = 4.7, *p* = 0.033). (4) Conclusions: At the time of exit from the first lockdown, compliance with social distancing measures was high, with Jewish, single and male Israelis more likely to adhere to the guidelines. We identified the populations at risk for non-adherence and associated factors, reporting for the first time the correlation between SoC and SRA. Further research is needed to assess the role of these factors in Jewish and Arab populations.

## 1. Introduction

As we enter the third year of the global pandemic, Omicron, the new variant of COVID-19 has shattered hopeful expectations and forced us to re-evaluate the prevention strategies that have been used so far. While new COVID-19 variants continue to emerge, the vaccine protection wanes, and many people around the world remain unvaccinated, social distancing prevails as one of the most effective protective strategies that limit the spread of the virus [1]. Due to the continuously high infection rates, social distancing should be and has been implemented as a legislative directive on a global scale [2]. However, turning the directive into reality is challenging, and we must obtain a deeper understanding of what drives people to adhere to the guidelines.

Since the outbreak of the pandemic, researchers have been vigorously studying the differences between people who adhere to social distancing and those who do not, or, in some cases, oppose the distancing [3,4,5,6]. The social-distance adherence continuum is revealed in literature as ranging from people who over-respond all the way to people who completely disregard the recommendation. On one end of the continuum is the “over-response”: the behavior of a distressed person who adopts an exaggerated form of social distancing that resembles self-isolation [7]. The “under-response” is understood as a disregard of the guidelines and of the dangers that come with COVID-19 [7].

To further our understanding of predictive indicators for adherence to the guidelines, we have reviewed relevant literature, including studies that have yet to be peer-reviewed. Due to the social desirability bias, the data collected must be analyzed with caution: a study that used the mobile data of 302 participants to track the differences between self-reported adherence to social distancing and real behavior found that the self-reports are consistently higher than the community average [8]. Among the factors identified as most highly associated with high adherence to the social distancing rules were one’s perception of their own risk, older age, recognition of one’s responsibility for their community, as well as the practical ability to do so, including the ability to work or study remotely [5,9,10]. One’s social circle appears to predict adherence to social distancing more than the individual’s own confidence in the guidelines [11]. A related predictor is the political standpoint of the individual. One of the studies on this matter investigated why the American conservatives are inclined to ignore the social-distancing guidelines [6]. The researchers found that adherence is related to the conservatives’ tendency to distrust science. Conservative people who reported trusting science were more inclined to socially distance themselves [6], and analytic thinkers tended to reject the conspiracy theories that have evolved regarding the COVID-19 outbreak [12]. A different study identified the impact of the politicization of the pandemic, including its representation in the liberal mainstream media, as what led the American conservatives to disregard the severity of the pandemic and doubt that social distancing can prevent its spread [13]. Further, boredom and self-control were identified as two important predictors of social-distancing adherence [14]. This high-powered cross-sectional study of 895 people revealed that individuals who scored high in the boredom trait found adherence to social distancing to be particularly challenging. The opposite was true for individuals with high self-control traits [14,15].

Research has been spreading out from merely looking into the traits that predict adherence to analyzing how public health promoters can call the public to action. One study found that messages to the public that emphasize one’s duty towards community and peers could be the key to promoting adherence [16]. Essentially, presenting social distancing as one’s moral duty adds an ethical dimension and meaning to the guidelines and frames the rejection of the adherence as immoral [17,18]. The urgency of such measures is clear, yet health promoters must use persuasion tools responsibly. Fear-based techniques have been shown to trigger emotional responses and are less effective than prosocial framing [19]. Eliciting fear is potentially dangerous as it can trigger extreme emotions and behaviors in populations predisposed to emotional disorders [19].

The current study, in conjunction with similar studies that are being conducted worldwide, was urgently needed because social distancing has been the crucial strategy for prevention of the collapse of medical systems [20]. Despite the calls for social distancing and legislation, the adherence has been far from perfect. A 2021 study conducted in London, England, revealed that 92.8% of 681 respondents did not adhere to all social-distancing rules, and 48.6% intentionally disregarded them [9]. The variability we are seeing in the adherence to the guidelines is one of the reasons why we need to identify the factors associated with adherence. Once identified, we will be more likely to address the issues successfully and lower the spread of COVID-19. Our role as health promoters is to inform policymakers on the most effective ways to prompt high adherence in their population and educate them on the populations that require special attention in this matter [21].

In the late 1970s, Aaron Antonovsky presented the theory of Salutogenesis, proposing that the way an individual copes with life is related to their health [22]. His Salutogenic model presents a paradigm shift that emphasizes the health end of the health-ease and dis-ease continuum—rather than the disease end [22]. The Sense of Coherence Scale (SoC) is the tool Antonovsky created in order to understand how one copes with life’s challenges. The 29-item and shortened 13-item SoC questionnaires measure three factors: comprehensibility, manageability, and meaningfulness of the perceived stressors [23]. As Hammond and Niedermann summarize, “the more a person is able to understand and integrate (comprehensibility), to handle (manageability) and to make sense (meaningfulness) of an experience or disease, the greater the individual’s potential to successfully cope with the situation or the disease” [24]. A large systematic review conducted in 2005 summarized 25 years of research and 458 studies on SoC [25], and found the scale to be valid, reliable, and cross-culturally applicable for measuring how people manage stressful situations and stay well.

One’s Sense of Coherence may be an essential source of resilience [23], defined as the capacity to adapt when faced with adversity and trauma [26]. Thus, SoC can partially explain the variability of the responses to stressors observed in society. The COVID-19 pandemic and the lockdowns have presented a unique, global challenge, stressor, and trauma, and tested individual, communal, and national resilience. While the national resilience declined due to the pandemic [27], many people were able to cope well thanks to their individual levels of resilience: 70% of the respondents in a large study in Italy displayed resilience, attributed by the authors mostly to the respondents’ trait resilience and conscientiousness [28]. An international study conducted simultaneously in seven countries in 2020 revealed that high SoC was correlated with better mental health during the pandemic and that it was mediated by perceived family support and trust in leaders and institutions [29]. These findings were confirmed by others, in relation to anxiety [30] and depression [31]. Thus, one’s ability to understand, manage and make sense of the crisis has been shown as crucial for their well-being. Following the findings regarding the trust in leaders and institutions as a mediator between SoC and mental health [29], we may theorize that high SoC will be associated with greater adherence to the imposed social distancing guidelines. While it has been proposed that strict adherence to social distancing may be a result of a negative fear-response to the crisis [32], a high SoC may result in the same outcome whilst maintaining the individual’s overall well-being. To the best of our knowledge, to date SoC has not been measured in relation to the adherence to guidelines.

Following the 3-component Salutogenic model, if an individual is able to comprehend the reason for social distancing (thanks to successful and trusting communication with leaders), manage the challenge it presents (thanks to their own and societal resources), and make meaning of it (see both the individual and the communal value of social distancing), we hypothesize that they will be more likely to adhere to the guidelines. We further hypothesize that higher perceptions of protection and danger will be associated with greater Self-reported Adherence (SRA), and that adherence and SoC will differ between people of different genders, nationalities, occupations, income, education, religion and religiosity and marital status. The hypothesized differences between representatives of the two main nationalities residing in Israel—Jews and Arabs—is based on previous studies that reported poorer health status in the Arab population and differences in health behaviors and well-being among the two groups [33]. Thus, the current study set out to collect and analyze data on the SRA to the social distancing guidelines and the factors associated with SRA, including socio-demographic factors, SoC, perceptions of protection and danger. Additionally, our goal was to identify the factors associated with SoC. The data were collected in Israel at the end of the first COVID-19 lockdown, in the spring and summer of 2020.

## 2. Materials and Methods

In Israel, the first case of COVID-19 was diagnosed on 21 February 2020. The social distancing regulations began on 11 March 2020 and escalated on 19 March 2020, when a national state of emergency was declared, followed by restrictions on movement: the population was restricted to a 100-m radius from home for any nonessential activities. The questionnaire used for this study was distributed during the gradual withdrawal from the first-wave lockdown, between 14 May and 22 June 2020. We conducted a cross-sectional survey, inviting the participants using the iPanel platform. iPanel is the largest panel in Israel and it strictly adheres to the European Society for Opinion and Marketing Research (ESOMAR) principles, while all results are filtered through multi-stage validation before final data analysis [34,35]. The study protocol was approved by the Ethics Committee (Institutional Review Board) of Tel Aviv University (#0001338-2).

The quantitative questionnaire consists of previously validated questionnaires and items related specifically to COVID-19. All questions and explanations were written both in Hebrew and Arabic for the respective Israeli populations. The independent variables included standard social-demographics questions used to assess gender, age, religion, marital status, income, as previously described.

The dependent variables were 7 measures of adherence to social distancing, according to a self-report. The social distancing questions were based on the rapidly developed ones in the International Tobacco Control Policy Evaluation (ITC) project surveys. The social distancing measures were defined as follows: avoiding public places, avoiding contact with non-cohabitating family or friends, maintaining a 2-m distance from others, coughing into a tissue or elbow, handwashing practice, style of greeting, adherence to the measures during holidays, and remaining within a 100 m radius from one’s home, per national guidelines. We created a social distancing scale (SDS) that is based on these 7 measures. We used a Likert scale, ranging from 1 (strongly disagree) to 5 (strongly agree). The outcome of the SDS ranges from 7 to 35, with higher scores indicating the highest level of adherence to social distancing measures. The validity of questionnaires was tested via Pearson’s correlation coefficient, examining linear relationships between consecutive variables. Reliability was tested via Cronbach’s alpha measure (alpha = 0.68).

We collected data on the following explanatory variables: sociodemographic characteristics, sleep, health status, perceptions of pandemic-related danger and protection (provided by the social distancing measures), Sense of Coherence (SoC), and subjective well-being. For the analysis of demographic characteristics, we performed independent t-tests for continuous variables, and chi-square tests for categorical variables. Independent t-tests and ANOVA were utilized for the analysis of variables associated with the sense of coherence. Linear regression was performed to assess the association between SRA and explanatory variables.

The Sense of Danger Scale (SDS) was used to assess the risk perceptions. The respondents were asked how endangered they felt by the spread of the virus, using a Likert-like scale ranging from 1 (not at all) to 5 (very much) [36]. Similar questions were used in other studies on the risk perceptions during the pandemic [37,38,39]. The perception of protection provided by the social distancing measures was quantified using questions relating both to the individual and the societal level (“how personally protected from the COVID-19 do you feel by the governmental guidelines regarding social distancing?”; “how successful do you think that the government-advised social distancing is at preventing the spread of the virus on a societal level?”). The possible answers regarding the perceived protective value ranged from not at all through somewhat to very much, with an additional “don’t know” option. The questions were based on the ITC Project surveys [40].

The Sense of Coherence was measured using the shortened, 13-item questionnaire developed by A. Antonovsky [23]. The questionnaire addresses three components of the model: comprehensibility, manageability, and meaningfulness of the perceived stressors. The response scale to each question ranges from 1 to 7. The total score of the questionnaire ranges from 13 to 91 points, with higher scores indicating a greater sense of coherence and greater subjective well-being (alpha = 0.84).

## 3. Results

In total, 807 respondents completed the survey. The response rate stood at 25% for the Jewish population and 10% for the Arabic population. Distributions of gender, age (18–70), and nationality (Arab 20%, Jewish 80%) matched the national distributions. Table 1 presents the socio-demographic characteristics of the study population by nationality and as a whole.

### 3.1. Compliance with the Seven Social Distancing Measures

Figure 1 presents the self-reported adherence to the social distancing guidelines. The majority (60%) of respondents reported complying with the 7 measures all or most of the time (avoiding public places: 72.3%, avoiding people from outside of one’s household: 65.1%, maintaining 2-m distance outside the home: 82.1%, coughing in elbow or tissue: 83.4%, handwashing: 69.6%, greeting without physical contact: 81.5%, staying within 100 m of home: 80.6%). Just under half (46.2%) reported staying within 100 m of the home all of the time. Self-Reported Adherence among Jews was slightly higher than among Arabs (Jews:Mean = 10.5, SD = 4.5; Arabs:Mean = 9.1, SD = 4.1, *p* < 0.001) and higher among males than females (Males:Mean = 10.8, SD = 4.7; Females:Mean = 9.7, SD = 4.1, *p* = 0.003).

### 3.2. Multivariate Analysis of Variables Associated wih Social Distancing Adherence

Using both univariate and multivariate regressions, we found 6 independent variables significantly correlated with SRA: Perception of Protection (*p* < 0.001), nationality (*p* < 0.001), Sense of Coherence (*p* < 0.042), gender (*p* < 0.003), relationship status (*p* < 0.033), and Perception of Danger (*p* < 0.005). Those without partners reported significantly higher levels of Self-Reported Adherence than those with partners. Men were more likely to adhere to the guidelines than women, as was the Jewish population as compared to the Arabic population. Age, religiosity, family income and education did not have a significant correlation with SRA. These results are presented in Table 2.

### 3.3. Sense of Coherence

The sample mean of SoC levels stood at 61.1 out of the possible 91 points (SD = 13.8, *p* < 0.001). Out of the 3 components that make up the Sense of Coherence, manageability ranked the highest. The analysis of SoC according to demographic factors revealed that income, marital status, age, profession, and education were associated with it. As presented in Table 3, married respondents had greater SoC than non-married individuals (63.6 next to 57.9, t = −5.90, *p* < 0.001). Higher income and education had a significant impact on SoC, with higher levels observed in college graduates as opposed to those with no college degrees (62.6 next to 59.8, t = 2.85, *p* = 0.005), and highest SoC levels in those with graduate degrees (65.4, F = 6.30, *p* < 0.001). However, people with elementary and lesser education had higher SoC levels than those with high school diplomas (62.7 next to 57.1), at a lever similar to those with some college education (62.9). Career soldiers had the highest levels of SoC of all professional groups examined (69.5, F = 9.11, *p* < 0.001), followed closely by retirees (68.4), while the lowest levels of SoC were observed in soldiers undergoing their mandatory military service (52.8, *p* < 0.001). Older respondents had greater SoC than younger ones (t = 7.80, *p* < 0.001).

## 4. Discussion

Our study presents valuable data on the factors associated with adherence to the social distancing guidelines imposed during the COVID-19 pandemic in Israel. Thanks to the proximity of the study population’s socio-demographic characteristics to the national ones in Israel, its findings are reliable and possibly generalizable, if limited by the cross-sectional design of the study. Despite its effectiveness, social distancing has been a challenge: as revealed in this study, the adherence in Israel has been less than ideal already at the early stages, with approximately 60% of the respondents complying with all seven social-distancing measures at three months into the pandemic, defined as: avoiding public places, avoiding people from outside of one’s household, maintaining a 2-m distance from others while outside, coughing into elbow or tissue, proper handwashing, greeting people without physical contact, and staying within 100 m of one’s home. Coughing into the elbow or tissue, maintaining a 2-m distance from others when outside, greeting without physical contact, and remaining within a 100-m radius from one’s home, were the most commonly adhered to guidelines, at least most of the time, in the population analyzed for this study. However, about a third of the respondents did not avoid public places or people from outside of their households or followed the recommended handwashing routines. Thus, special attention needs to be devoted to promoting these aspects of the social distancing guidelines.

The analysis of collected data revealed six variables that were significantly correlated with one’s adherence to the social distancing guidelines: perception of protection, perception of danger, Sense of Coherence (SoC), gender, relationship status, and nationality. The perceptions of protection and danger depend on one’s understanding of the pandemic and the value of prevention measures, in addition to having trust in science and the government. If this trust erodes or the public health message does not reach the public, individuals are less likely to adhere to the guidelines, as was shown in large studies conducted in the United States [6,12]. In our study, greater perceptions of protection and danger were correlated with increased adherence to guidelines. Thus, we may hypothesize that adherence could increase if the perceptions of protection and danger are strengthened, possibly through successful and multi-faceted communication strategies between the leaders and the public. Age and sleep did not have a significant correlation with SRA. In other studies on this subject, older age was correlated with greater SRA [9], yet a multivariate analysis performed here showed that its effect became insignificant when other explanatory variables came into play.

In our study, men, single people, and Jews were more likely to adhere to the guidelines than their counterparts. It has been shown that one’s social circle predicts compliance with social distancing measures more than the individual’s confidence in the policy [11], explaining the correlation between the relationship status and compliance. Living with a partner might introduce opposing views on social distancing and widen the social circle. The finding that men in Israel were more likely to report adherence to guidelines stands in contrast with what was reported globally. A large study that combined data from eight countries showed a trend in the opposite direction, with women perceiving the pandemic as much more serious a threat than men and being more likely to adhere to the social-distancing advice [41]. Among the factors that might have contributed to this finding may be the child-rearing, cooking, and housekeeping burden that falls predominantly on women [42] and requires a greater reliance on others, inhibiting women’s ability to fully adhere to the guidelines. In the Israeli society, child-rearing is to a large extent a communal effort, and relies on the help of family members, including non-cohabiting ones like grandparents [43,44]. Further, if women are more responsible for grocery shopping than men, as is the case in the United States but has not been measured in Israel [45], they may be less likely to fully adhere to the social-distancing guidelines. Our finding that Jews were more likely to adhere to the guidelines is supported by previous studies that showed a greater toll of the pandemic in the Arabic population of Israel than in the country’s Jewish residents [46]. This outcome has been attributed to a myriad of factors, including their lower socio-economic status and education, lower than average trust in the government, unequal distribution of resources within the Israeli society, societal traditions and religious customs, and difficulty in maintaining health behaviors after the initial success of the first lockdown in Israel [46].

To the best of our knowledge, this is the only study conducted to date that examined the influence of one’s SoC on self-reported adherence to public health guidelines during the COVID-19 pandemic. It has been shown that SoC impacts one’s health status and chronic illness severity through its correlation with adherence to treatment guidelines: this mechanism has been described in people with Chronic Obstructive Pulmonary Disease (COPD) [47], Human Immunodeficiency Virus (HIV) [48], and Inflammatory Bowel Disease (IBD) [49]. Our study confirms this finding in a population facing the threat of an illness rather than the illness itself: people with higher SoC were more likely to adhere to the national guidelines that were imposed in order to minimize the spread of the virus.

It is generally accepted that SoC is a significant predictor of Quality of Life (QoL), with higher SoC levels correlated with better QoL [50]. Our analysis supports this connection, revealing that the SoC levels were highest in people with higher incomes and college degrees. Age was correlated with one’s SoC, confirming previously reported findings on the continuous increase of SoC into old age [51]. One’s profession was associated with their SoC, with an interesting finding regarding the military: career soldiers had the highest SoC while the soldiers undergoing compulsory service had the lowest levels. This correlation might be in part related to the age of the soldiers—the Israeli Army drafts 18- and 19-year-olds—but it may also be connected to the young soldiers’ perceived lack of control over their lives. The locus of control and SoC are related constructs [52], and the COVID-19 pandemic appears to impact most negatively the younger population through decreasing their internal locus of control and SoC [53]. On the other hand, religiosity, gender, place of birth and the number of people in one’s household did not have a significant association with the respondents’ SoC.

Our findings support the use of the SoC model for successful communication between policymakers and the public. Coherent, precise, and straightforward communication will enable the residents to comprehend the reasons for adhering to the guidelines and comply with the social distancing policy, thus fulfilling the first element of the SoC model, comprehensibility. Further, the policymakers must make the measures manageable for people of all backgrounds, both through publicizing simple and attainable strategies for social distancing (such as tips on hygiene or alternative greetings) and through providing the citizens with resources and practical support they need (such as financial support, universal access to the Internet, telemedicine, well-organized online learning, online support groups, and more). Lastly, it is crucial to pinpoint the practical and ethical value of adhering to the guidelines, both on an individual and a communal level, thus fulfilling the final element of the SoC model, meaningfulness. Further research is needed on the factors that are associated with SoC and adherence to guidelines, including one’s trust in science and the government, as well as the differences in the Jewish and Arab populations in this context.

The study measures self-reported adherence to the social distancing guidelines. Thus, the social desirability bias comes into play, and the data collected must be interpreted with caution. It has been found that self-reports on adherence to social-distancing guidelines are consistently higher than the community average [8]. However, the study provides an important approximation of the adherence to guidelines, the reliability of which was strengthened by the anonymity of the data collection. Further, the bias does not negate this study’s findings regarding the factors associated with adherence. The cross-sectional nature of the study does not enable us to make causal inferences but only to identify correlations, some of which may have been muddied by unidentified confounders.

## 5. Conclusions

Compliance with the social distancing measures was generally high among Israeli respondents at the time of exit from the first lockdown. However, even at that time, close to a third of the respondents did not engage in the recommended handwashing routines, and nearly 20 percent did not maintain a 2-m distance when outside of the house. To contain COVID-19 in the absence of a lockdown, policies and education are needed to encourage ongoing, high compliance with social distancing. Identifying the populations least likely to adhere to the social distancing guidelines is crucial for health promoters and policymakers. We identified the populations at risk of non-adherence and some associated factors, beliefs and constructs. Women, Arabs, and people in relationships were identified as less likely to comply than their counterparts. Further, we showed the significant correlation of the perceptions of protection and danger with adherence, and reported for the first time its association with the Sense of Coherence. Thanks to identifying these factors, we may better fulfill our role as health promoters and inform the policymakers on effective ways to promote prevention and limit the spread of the disease.

## Figures and Tables

**Figure 1 ijerph-19-08041-f001:**
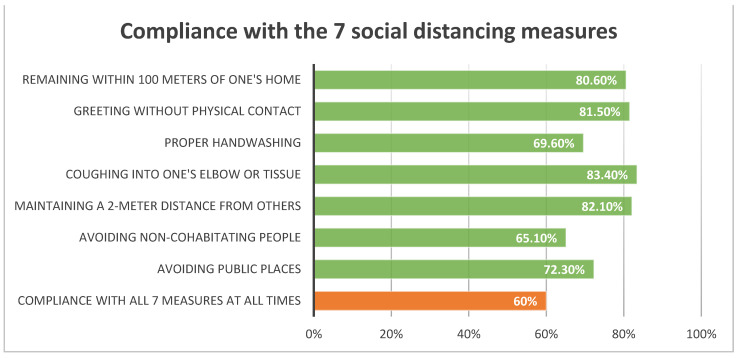
Compliance with the 7 social distancing measures.

**Table 1 ijerph-19-08041-t001:** Demographic statistics by nationality.

Characteristic		Study Population Total N, (%) or Mean (SD)	Jewish N (%) or Mean (SD)	Arabic N (%) or Mean (SD)	*p* Value
Age	[18.0, 89.0]	39.3 (14.7)	40.4 (14.8)	35.0 (13.7)	<0.001
Number of People at Home	[1.0, 13.0]	3.8 (1.8)	3.7 (1.8)	4.5 (2.0)	<0.001
Family Income	Below average	321 (47.0)	188 (35.2)	18 (12.1)	<0.001
	Average	156 (22.8)	128 (24.0)	28 (18.8)	
	Above average	206 (30.2)	218 (40.8)	103 (69.1)	
Gender	Male	404 (50.1)	317 (49.2)	87 (53.4)	0.344
	Female	403 (49.9)	327 (50.8)	76 (46.6)	
Marital Status	Without partner	357 (44.2)	284 (44.1)	73 (44.8)	0.875
	With partner	450 (55.8)	360 (55.9)	90 (55.2)	
Religious	No	654 (81.2)	535 (83.1)	119 (73.9)	0.008
	Yes	151 (18.8)	109 (16.9)	42 (26.1)	
Religiosity	Secular	375 (46.6)	335 (52.0)	40 (24.8)	<0.001
	Traditional	279 (34.7)	200 (31.1)	79 (49.1)	
	Religious	131 (16.3)	91 (14.1)	40 (24.8)	
	Ultra-religious	20 (2.5)	18 (2.8)	2 (1.2)	
Education	Elementary or less	7 (0.9)	5 (0.8)	2 (1.2)	<0.001
	High school, no diploma	76 (9.4)	56 (8.7)	20 (12.3)	
	High school with diploma	172 (21.3)	145 (22.5)	27 (16.6)	
	College with no degree	175 (21.7)	144 (22.4)	31 (19.0)	
	First degree	272 (33.7)	204 (31.7)	68 (41.7)	
	Second degree and higher	105 (13.0)	90 (14.0)	15 (9.2)	
Birthplace	Israel	653 (88.0)	555 (86.2)	98 (100.0)	<0.001
	Abroad	89 (12.0)	89 (13.8)	0 (0.0)	
District	Jerusalem	64 (7.9)	59 (9.2)	5 (3.1)	<0.001
	North	140 (17.3)	63 (9.8)	77 (47.2)	
	Haifa	109 (13.5)	78 (12.1)	31 (19.0)	
	Center	201 (24.9)	181 (28.1)	20 (12.3)	
	Tel Aviv	145 (18.0)	139 (21.6)	6 (3.7)	
	South	117 (14.5)	93 (14.4)	24 (14.7)	
	Judea and Samaria	31 (3.8)	31 (4.8)	0 (0.0)	
Education	No	430 (53.3)	294 (45.7)	83 (50.9)	0.228
	College degree	377 (46.7)	350 (54.3)	80 (49.1)	
Employment	Mandatory soldier	84 (10.4)	19 (3.0)	65 (39.9)	<0.001
	Professional soldier	6 (0.7)	4 (0.6)	2 (1.2)	
	Higher education student	68 (8.4)	51 (7.9)	17 (10.4)	
	Part time employee	61 (7.6)	53 (8.2)	8 (4.9)	
	Full time employee	366 (45.4)	320 (49.7)	46 (28.2)	
	Independent	54 (6.7)	49 (7.6)	5 (3.1)	
	Retired with pension	43 (5.3)	41 (6.4)	2 (1.2)	
	Unemployed	77 (9.5)	62 (9.6)	15 (9.2)	
	Vacation without pay	48 (5.9)	45 (7.0)	3 (1.8)	

**Table 2 ijerph-19-08041-t002:** Multivariate analysis of variables associated with social distancing adherence.

Variable	Levels	Mean (SD)	Univariable		Multivariate	
			Coefficient	*p*	Coefficient	*p*
Perception of Protection	[1, 6]	10.2 (4.5)	0.84 (0.64, 1.03)	<0.001	0.69 (0.45, 0.93)	<0.001
Perception of Danger	No	10.7 (4.6)		<0.001		0.005
	Yes	9.4 (4.1)	−1.37 (−2.01, −0.72)		−1.10 (−1.86, −0.34)	
Sense of Coherence	[18, 91]	10.2 (4.5)	−0.03 (−0.05, −0.01)	0.016	−0.03 (−0.06, −0.00)	0.042
Gender	Male	10.8 (4.7)		<0.001		0.003
	Female	9.7 (4.1)	−1.13 (−1.75, −0.52)		−1.09 (−1.80, −0.38)	
Nationality	Arab	9.1 (4.1)		0.001		<0.001
	Jew	10.5 (4.5)	1.37 (0.60, 2.14)		1.98 (1.05, 2.91)	
Relationship Status	Partnered	9.7 (4.2)		<0.001		0.033
	Not partnered	10.9 (4.7)	1.18 (0.56, 1.80)		0.92 (0.07, 1.78)	
Religious	Yes No	10.1 (4.8) 10.3 (4.8)	0.18 (−0.63, 0.98)	0.667	−0.38 (−1.31, 0.54)	0.418
Education	college below	10.1 (4.4) 10.3 (4.5)	0.27 (−0.35, 0.89)	0.397	−0.19 (−0.91, 0.53)	0.602
Family income	Below average Average Above average	10.0 (4.6) 10.7 (4.5) 10.4 (4.2)	0.71 (−0.16, 1.57) 0.37 (−0.42, 1.15)	0.257	0.13 (−0.79, 1.05) 0.05 (−0.83, 0.93)	0.962
Age	[18, 70]	10.2 (4.5)	−0.04 (−0.06, −0.02)	<0.001	−0.03 (−0.06, 0.00)	0.094

**Table 3 ijerph-19-08041-t003:** Demographic characteristics associated with sense of coherence.

	Level	Value	N (%)	Value (t/F)	*p* Value
Gender	Male	61.9 (13.0)	404 (50.1)	t = 1.73	0.083
	Female	60.3 (14.5)	403 (49.9)		
Marital Status	No	57.9 (13.9)	357 (44.2)	t = −5.90	<0.001
	Yes	63.6 (13.1)	450 (55.8)		
Religious	No	60.7 (13.7)	654 (81.2)	t = −1.76	0.075
	Yes	62.9 (14.0)	151 (18.8)		
Number of People in Household	[1.0, 13.0]	61.1 (13.8)	806 (100.0)	t = −0.42	0.676
College Education	College degree	62.6 (13.2)	377 (46.7)	t = 2.85	0.005
	None	59.8 (14.1)	430 (53.3)		
Family Income	Above average	64.1 (13.6)	206 (30.2)	F = 10.04	<0.001
	Average	61.8 (13.6)	156 (22.8)		
	Below average	58.7 (13.6)	321 (47.0)		
Age	[18.0, 89.0]	61.1 (13.8)	807 (100.0)	t = 7.80	<0.001
Birthplace	Israeli born	61.5 (13.6)	653 (88.0)	t = 3.79	0.052
	Born abroad	64.4 (13.4)	89 (12.0)		
District	Jerusalem	64.3 (11.9)	64 (7.9)	F = 2.21	0.040
	North	59.3 (13.5)	140 (17.3)		
	Haifa	62.3 (13.7)	109 (13.5)		
	Center	61.2 (14.3)	201 (24.9)		
	Tel Aviv	60.8 (14.0)	145 (18.0)		
	South	59.3 (13.4)	117 (14.5)		
	Judea and Samaria	66.2 (14.3)	31 (3.8)		
Religiosity	Secular	61.4 (13.9)	375 (46.6)	F = 1.73	0.159
	Traditional	59.8 (13.4)	279 (34.7)		
	Religious	63.0 (13.9)	131 (16.3)		
	Ultra-religious	62.8 (15.1)	20 (2.5)		
Education	Elementary or less	62.7 (17.8)	7 (0.9)	F = 6.30	<0.001
	High school no diploma	58.6 (14.2)	76 (9.4)		
	High school with diploma	57.1 (14.4)	172 (21.3)		
	High school, no college degree	62.9 (13.1)	175 (21.7)		
	First degree	61.5 (13.2)	272 (33.7)		
	Second degree and higher	65.4 (12.6)	105 (13.0)		
Profession	Mandatory soldier	52.8 (13.1)	84 (10.4)	F = 9.11	<0.001
	Career soldier	69.5 (11.7)	6 (0.7)		
	Student—academia	57.9 (12.4)	68 (8.4)		
	Part time employee	62.4 (12.4)	61 (7.6)		
	Full time employee	63.1 (13.7)	366 (45.4)		
	Independent	64.2 (12.6)	54 (6.7)		
	Retired with pension	68.4 (13.4)	43 (5.3)		
	Unemployed	57.3 (13.8)	77 (9.5)		
	Vacation without pay	58.3 (11.7)	48 (5.9)		

## Data Availability

The data presented in this study are available on request from the corresponding author.

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
