# Peer review of "The Association between the Sense of Coherence and the Self-Reported Adherence to Guidelines during the First Months of the COVID-19 Pandemic in Israel"

_ijerph, 2022, doi:10.3390/ijerph19138041_

Round 1

Reviewer 1 Report

How was the income level of the respondents determined?

What was the response rate on your questionnaire?

How were the participants invited?

Author Response

Thank you for taking time to read our manuscript and submit your comments. Please find our responses to your questions below.

Question 1:

How was the income level of the respondents determined?

Thank you for the question. The income level of the respondents was determined by the respondents themselves. They were asked to identify their income on a scale that included below average, average and above average income. They were provided with the most recent average monthly income for reference.

This information appears in the original manuscript (lines 152-153).

Question 2:

What was the response rate on your questionnaire?

The Response rate of IPANEL digital survey was between approximately 25% (Jewish population) to 10% (Arab population).

This information was added to the manuscript, in the Results section (lines 188-9).

Question 3:

How were the participants invited?

Thank you for asking. They were invited using the iPanel platform, the largest panel in Israel. We have added a clarification.

The following information was added to the manuscript, in the Methods section (line 142): „We conducted a cross-sectional survey, inviting using the participants using the iPanel platform. iPanel is the largest panel in Israel and it strictly adheres to the European Society for Opinion and Marketing Research (ESOMAR) principles…:

Reviewer 2 Report

Authors:

Thank you for the opportunity to review the article” The association between the Sense of Coherence and the self-2 reported adherence to guidelines during the first months of the 3 COVID-19 pandemic in Israel”

Overall, it is a good manuscript, however there are a few specific issues to be addressed as follows:

1) Abstract:

  • You mentioned in the abstract and method section that you collected data about lifestyle (smoking, sleep, and physical activity) and health status. However, you just report sleep data, could you report the other data?
  • The abstract exceeds the maximum of 200 words allowed; the authors could summarize it.

2) Introduction

  • It is okay. The introduction briefly describes the study and purpose. It also includes the current state of the research field providing sufficient background, and the main aim of the work.

3) Methods

  • This section could be improved, using subheadings (Study design, participants, measurements, procedure, and statistical analysis).
  • You mentioned in the abstract and method section that you collected data about lifestyle (smoking, sleep, and physical activity) and health status. However, you just report sleep data.
  • Could you mention what kind of analysis you did in the Demographic statistics by nationality?

4) Results

  • You mentioned in the abstract and method section that you collected data about lifestyle (smoking, sleep, and physical activity) and health status. However, you just report sleep data, could you report the other data?
  • In table 1, why did you put NS instead of the p value? You can put the p value in each characteristic and add * in significance p values, then in the foot table add a note with *p<.001

Author Response

Thank you for reviewing our paper and submitting your comments. We address each of your remarks in this document and we are re-submitting the edited manuscript.

1) Abstract:

  • You mentioned in the abstract and method section that you collected data about lifestyle (smoking, sleep, and physical activity) and health status. However, you just report sleep data, could you report the other data?

Thank you for reading our article closely and noticing it. An analysis of lifestyle variables exceeds the scope of this article. We have corrected the methods to include variables that were analyzed in this study.

  • The abstract exceeds the maximum of 200 words allowed; the authors could summarize it.

 Thank you. We have shortened the abstract slightly following your suggestion.

2) Introduction

  • It is okay. The introduction briefly describes the study and purpose. It also includes the current state of the research field providing sufficient background, and the main aim of the work.

Thank you for your comment.

3) Methods

  • This section could be improved, using subheadings (Study design, participants, measurements, procedure, and statistical analysis).

Thank you for noticing. We have followed the format required by the journal (through the provided template), and it does not allow for subheadings.

  • You mentioned in the abstract and method section that you collected data about lifestyle (smoking, sleep, and physical activity) and health status. However, you just report sleep data.

Thank you for reading our article closely and noticing it. An analysis of lifestyle variables exceeds the scope of this article. We have corrected the methods to include variables that were analyzed in this study.

  • Could you mention what kind of analysis you did in the Demographic statistics by nationality?

Thank you for asking. For continuous variables, we performed independent t-tests and for categorical variables, chi-square test.  We have added this information to the manuscript (lines 168-169).

4) Results

  • You mentioned in the abstract and method section that you collected data about lifestyle (smoking, sleep, and physical activity) and health status. However, you just report sleep data, could you report the other data?

Thank you for reading our article closely and noticing it. An analysis of lifestyle variables exceeds the scope of this article. We have corrected the methods to include variables that were analyzed in this study.

  • In table 1, why did you put NS instead of the p value? You can put the p value in each characteristic and add * in significance p values, then in the foot table add a note with *p<.001

Thank you for bring this to our attention. We have completed the table with the p values for each characteristic (see table 1 in manuscript).

Reviewer 3 Report

I consider this article very interesting.  Yet, there are some questions to be answered:

  1. Are there any other studies using the SoC measurement related to COVID-19?
  2. Is there any possibility that SRA bias correlated to demographic features? For example, the difference of SRA between Jews and Arabs was slight. And how to exclude the measurement bias or self-report difference/reluctancy among population groups?
  3. How to exclude other factors' influences? For example, the trust in science and the government, how do they interact with SoC or SRA? 
  4. Would be the results in Table 2 robust if controlling the demographic characteristics?

Author Response

Thank you for reading our paper and submitting your questions and comments. We were happy to hear that you found interesting. We address each of your remarks in this document and we are re-submitting the edited manuscript.

  1. Are there any other studies using the SoC measurement related to COVID-19?

Thank you for your question. As stated in the introduction, the Sense of Coherence measurement was used in other studies conducted during the pandemic:

An international study conducted simultaneously in seven countries in 2020 revealed that high SoC was correlated with better mental health during the pandemic and that it was mediated by perceived family support and trust in leaders and institutions [26]. These findings were confirmed by others, in relation to anxiety [27] and depression [28]. Thus, one’s ability to understand, manage and make sense of the crisis has been shown as crucial for their well-being. Following the findings regarding the trust in leaders and institutions as a mediator between SoC and mental health [26], we may theorize that high SoC will be associated with greater adherence to the imposed social distancing guidelines. While it has been proposed that strict adherence to social distancing may be a result of a negative fear-response to the crisis [29], a high SoC may result in the same outcome whilst maintaining the individual’s overall well-being. Following the 3-component Salutogenic model, if an individual is able to comprehend the reason for social distancing (thanks to successful and trusting communication with leaders), manage the challenge it presents (thanks to their own and societal resources), and make meaning of it (see both the individual and the communal value of social distancing), we hypothesize that they will be more likely to adhere to the guidelines.

[26]: Mana, A.; Bauer, G.F.; Meier Magistretti, C.; Sardu, C.; Juvinyà-Canal, D.; Hardy, L.J.; Catz, O.; Tušl, M.; Sagy, S. Order out of chaos: Sense of coherence and the mediating role of coping resources in explaining mental health during COVID-19 in 7 countries. SSM Ment. Health 2021, 1, 100001, doi:10.1016/j.ssmmh.2021.100001.

[27]: Leung, A.Y.M.; Parial, L.L.; Tolabing, M.C.; Sim, T.; Mo, P.; Okan, O.; Dadaczynski, K. Sense of coherence mediates the relationship between digital health literacy and anxiety about the future in aging population during the COVID-19 pandemic: a path analysis. Aging Ment. Health 2021, 1–10, doi:10.1080/13607863.2020.1870206.

[28]: Schmuck, J.; Hiebel, N.; Rabe, M.; Schneider, J.; Erim, Y.; Morawa, E.; Jerg-Bretzke, L.; Beschoner, P.; Albus, C.; Hannemann, J.; Weidner, K.; Steudte-Schmiedgen, S.; Radbruch, L.; Brunsch, H.; Geiser, F. Sense of coherence, social support and religiosity as resources for medical personnel during the COVID-19 pandemic: A web-based survey among 4324 health care workers within the German Network University Medicine. PLoS ONE 2021, 16, e0255211, doi:10.1371/journal.pone.0255211.

[29]: Barni, D.; Danioni, F.; Canzi, E.; Ferrari, L.; Ranieri, S.; Lanz, M.; Iafrate, R.; Regalia, C.; Rosnati, R. Facing the COVID-19 Pandemic: The Role of Sense of Coherence. Front. Psychol. 2020, 11, 578440, doi:10.3389/fpsyg.2020.578440.

  1. Is there any possibility that SRA bias correlated to demographic features? For example, the difference of SRA between Jews and Arabs was slight. And how to exclude the measurement bias or self-report difference/reluctancy among population groups?

Thank you for your comment. As you noted, the social desirability bias does come into play in the current study, and the data collected must be interpreted with caution (as we state in the discussion chapter). It has been found that self-reports on adherence to social-distancing guidelines are consistently higher than the community average [3]. However, the study provides an important approximation of the adherence to guidelines, the reliability of which was strengthened by the anonymity of the data collection. Further, the bias does not negate this study’s findings regarding the predictors for adherence. We controlled for the demographic features in our multivariate analyses, and while there exists a possibility of a bias related to demographic features, we attempted to overcome these as much as possible by recruiting a large sample and collecting data anonymously. As reported by Saban et al. in 2021, the incidence of COVID-19 in Israel varied between the Jewish and Arabic citizens, with less hospitalizations in the Arabic population during the first wave [39]. This was accredited by the authors in part to high compliance with public health measures during the first wave [39]. The situation changed in the following waves possibly in part due to greater distrust in the government (a variable that was not measured in the current study), as well as generally lower level of income and education (two variables that were measured in the study and included in the analyses).

[3]: Gollwitzer, A.; McLoughlin, K.; Martel, C.; Marshall, J.; Höhs, J.M.; Bargh, J.A. Linking Self-Reported Social Distancing to Real-World Behavior During the COVID-19 Pandemic. Soc. Psychol. Personal. Sci. 2021, 194855062110181, doi:10.1177/19485506211018132.

[39]: Saban, M.; Myers, V.; Peretz, G.; Avni, S.; Wilf-Miron, R. COVID-19 morbidity in an ethnic minority: changes during the first year of the pandemic. Public Health 2021, 198, 238–244, doi:10.1016/j.puhe.2021.07.018.

  1. How to exclude other factors' influences? For example, the trust in science and the government, how do they interact with SoC or SRA? 

Unfortunately, the current study did not include a measurement of one’s trust in science and the government. In the last two years, many researchers noted the impact of trust on adherence to the pandemic guidelines (as well as the impact of one’s political beliefs on trust, and thus adherence). However, to the best of our knowledge, no study addressed directly the correlations between Sense of Coherence and trust. As we report in the introduction:

A related predictor (of SRA) is the political standpoint of the individual. One of the studies on this matter investigated why the American conservatives are inclined to ignore the social-distancing guidelines [8]. The researchers found that adherence is related to the conservatives’ tendency to distrust science. Conservative people who reported trusting science were more inclined to socially distance themselves [8], and analytic thinkers tended to reject the conspiracy theories that have evolved regarding the COVID-19 outbreak [9]. A different study identified the impact of the politicization of the pandemic, including its representation in the liberal mainstream media, as what led the American conservatives to disregard the severity of the pandemic and doubt that social distancing can prevent its spread [10].

It appears plausible that one’s SoC would be related to their trust in science and the government, and this could be true independently of one’s education level (in our study, the SoC varied between different levels of education and there was no trend). It is an interesting direction for future research, and we added this idea to the final paragraphs of our article. 

Lines 318-320:

Further research is needed on the factors that are associated with SoC and adherence to guidelines, including one’s trust in science and the government.

  1. Would be the results in Table 2 robust if controlling the demographic characteristics?

Thank you for asking. The multivariable analysis results of which are presented in table 2, included the following demographic characteristics: age, relationship status, nationality and gender.

As stated in the results chapter, we found 6 independent variables significantly correlated with self-reported adherence to the social distancing guidelines: Perception of Protection (p < .001), nationality (p < .001), Sense of Coherence (p < .042), gender (p < .003), relationship status (p < .033), and Perception of Danger (p < .005). Those without partners reported significantly higher levels of Self-Reported Adherence than those with partners. Men were more likely to adhere to the guidelines than women, as was the Jewish population as compared to the Arabic population.

Reviewer 4 Report

The authors tried to evaluate the factors that affect sense of coherence and adherence to the social distancing guidelines during the COVID-19 lockdown in Israeli adults.
The writing is generally clear and the study topic can be of interest to the readers.
However, I have some concerns about the description of the statistical methods and the interpretation of the results.

Major concerns:
- Why does the author provide demographic statistics specifically based on nationality at Table 1? If the proportion of the two nationalities match that of the country's general distribution, I do not see any point in providing demographic information per nationality. If this was one of the important hypothesis, please elaborate on the introduction section.
- What statistical test does the P value at Table 1 represent? There is no description of what statistical methods are used to test the demographic characteristics.
- What does the 807(100.0) mean at the second column of the 'Age' row of the Table 1? The column header indicates that it is the mean value of a continuous variable, but I believe that the mean age cannot be 807. Please revise the Table to provide more concise information.
- Figure 1 is not referred to in any paragraph of the manuscript.
- Please provide effect sizes or statistical values (t/F) along with the P values
- Why did the authors employ both univariate and multivariate analysis? What kind of differences do they have? Please elaborate.
- The p values of 'Education' and 'Employment' are significant for some statistical test (which I do not know what test is performed as mentioned above). Does this mean that the collected samples are somewhat biased? If so, shouldn't these factors be corrected for the following statistical tests as confounds?
- Please specify in the methods section paragraph about which statistical tests were performed for the Table 3. Just providing t or F values seems insufficient.
- The authors report many significant results and interpret that these factors are all directly related to the target variable (Adherence and SoC). However, there is a high possibility that only a few of the significant results are actually (or causally) related to Adherence or SoC, while others are indirectly correlated (or confounded). The authors should be aware of this and provide sufficient interpretation for the possible confounding.
- First paragraph of the Discussion section seems to be already discussed in the Introduction section. Please consider removing the overlapped part.
- The authors insist that adherence to the social distancing guidelines could increase if the perceptions of protection and danger are strengthened. However, it should be assumed that there are causal relation between the sense of protection/danger and the adherence, while the statistical results in this paper only supports correlation between the two variables. The sentence at line 260-3 should be toned down.
- The authors interpret the reason for the lower adherence among women, which was different from previous research results, for the child-rearing/cooking/housekeeping burden. For this statement to be true, the housekeeping burden should be specific to the country or culture that of the subject population. Please support evidences on this interpretation that gender roles have affected the results to be different from other previous studies.
- Citations are generally missing to statements that require supportive evidences. Please provide appropriate references thoroughly, especially for the introduction section.

Minor concerns:
- Abbreviation SoC (line 18) is not defined before.
- individual' -> individual's (line 60)
- based on the from -> based on (line 155)
- Please do not start the sentence with numbers (line 194)
- Please be consistent on writing the p/P value.
- Please be consistent on whitespacing around the (in)equality notation.

Author Response

We appreciate you taking time to thoroughly read and examine our work. Thank you for reading our article and submitting your questions and comments. They have enabled to improve our draft. We are happy to hear that you found our writing clear and that you think that the topic can be of interest to readers. We are attaching a response to your comments regarding the description of the statistical methods and the interpretation of the results below.

Minor concerns:
- Abbreviation SoC (line 18) is not defined before.
- individual' -> individual's (line 60)
- based on the from -> based on (line 155)
- Please do not start the sentence with numbers (line 194)
- Please be consistent on writing the p/P value.
- Please be consistent on whitespacing around the (in)equality notation

Thank you for paying close attention to detail and noting these errors. All of these have been corrected in the manuscript.

Major concerns:

- Why does the author provide demographic statistics specifically based on nationality at Table 1? If the proportion of the two nationalities match that of the country's general distribution, I do not see any point in providing demographic information per nationality. If this was one of the important hypothesis, please elaborate on the introduction section.

Thank you for asking. While, as you noted, the proportion of the two nationalities in the study matched Israel’s general distribution, we wanted to compare the the behaviors and the characteristics of the two groups. Jewish and Arabic respondents differed in the characteristics, including age, religiosity, educational level, employment, income, number of cohabitants, and place of residence. We collected information on a variety of possibly related variables, and one’s ethnic and cultural background is strongly related to health-related behaviors. Indeed, we have found that Jews were more likely to adhere to the guidelines. As we state in the discussion, our finding that Jews were more likely to adhere to the guidelines is supported by previous studies that showed a greater toll of the pandemic in the Arabic population of Israel than in the country’s Jewish residents [39]. This outcome has been attributed to a myriad of factors, including their lower socio-economic status and education, lower than average trust in the government, unequal distribution of resources within the Israeli society, societal traditions and religious customs, and difficulty in maintaining health behaviors after the initial success of the first lockdown in Israel [39]. 

- What statistical test does the P value at Table 1 represent? There is no description of what statistical methods are used to test the demographic characteristics.

Thank you for asking. For continuous variables, we performed independent t-tests and for categorical variables, chi-square test.  We have added this information to the manuscript (lines 168-169).

- What does the 807(100.0) mean at the second column of the 'Age' row of the Table 1? The column header indicates that it is the mean value of a continuous variable, but I believe that the mean age cannot be 807. Please revise the Table to provide more concise information.

Thank you for the bringing our attention to it. 807 was the total number of respondents. We edited the table to include the actual mean age and number of people at home for the total study population (39.3 and 3.8, respectively).  

- Figure 1 is not referred to in any paragraph of the manuscript.

Thank you for noticing. We have added a reference to it in the manuscript (line 200).

- Please provide effect sizes or statistical values (t/F) along with the P values

We have updated the manuscript to include those.

- Why did the authors employ both univariate and multivariate analysis? What kind of differences do they have? Please elaborate.

Comparing the univariable and multivariable analysis, we can see that age was statistically correlated with social distancing, however it didn't remain significant in the multivariable model. All other variables were statistically significant in both univariable and multivariable models. Meaning that when controlling for all other variables these variables still stay significant and explain some of the social distancing behaviors.

- The p values of 'Education' and 'Employment' are significant for some statistical test (which I do not know what test is performed as mentioned above). Does this mean that the collected samples are somewhat biased? If so, shouldn't these factors be corrected for the following statistical tests as confounds?

Using the multivariable analysis, we have controlled for these variables.

- Please specify in the methods section paragraph about which statistical tests were performed for the Table 3. Just providing t or F values seems insufficient.

Thank you for bring our attention to this. We updated the manuscript and clarified that independent t-tests and ANOVA were performed. (lines 168-171).

- The authors report many significant results and interpret that these factors are all directly related to the target variable (Adherence and SoC). However, there is a high possibility that only a few of the significant results are actually (or causally) related to Adherence or SoC, while others are indirectly correlated (or confounded). The authors should be aware of this and provide sufficient interpretation for the possible confounding.

Thank you for the comment. We are aware that due to the study’s cross-sectional nature, it is difficult to make causal inferences. At no point in our discussion have we implied causation, and after careful re-reading of the manuscript we strengthened this point:

Lines 238-240 (beginning of Discussion):

. Thanks to the proximity of the study population’s socio-demographic characteristics to the national ones in Israel, its findings are reliable and possibly generalizable, if limited by the cross-sectional design of the study.

Lines 330-332 (end of Discussion):

Yet, the cross-sectional nature of the study does not enable us to make causal inferences but only to identify correlations, some of which may have muddied by unidentified confounders.

- First paragraph of the Discussion section seems to be already discussed in the Introduction section. Please consider removing the overlapped part.

Thank you for bringing our attention to the repetitions present in the paper. We have shortened the first paragraph of the discussion following your suggestion, and it now focuses solely on the results of the current study.

- The authors insist that adherence to the social distancing guidelines could increase if the perceptions of protection and danger are strengthened. However, it should be assumed that there are causal relation between the sense of protection/danger and the adherence, while the statistical results in this paper only supports correlation between the two variables. The sentence at line 260-3 should be toned down.

Thank you for your very important comment regarding the limitations of our findings. We have toned down the sentence following your suggestion:

 “In our study, greater perceptions of protection and danger were correlated with increased adherence to guidelines. Thus, we may hypothesize that adherence could increase if the perceptions of protection and danger are strengthened, possibly through successful and multi-faceted communication strategies between the leaders and the public.” (lines 261-165)

Further, since the submission of the manuscript, a new study on this matter was published: Risk perceptions and COVID-19 protective behaviors: A two-wave longitudinal study of epidemic and post-epidemic periods https://doi.org/10.1016/j.socscimed.2022.114949

- The authors interpret the reason for the lower adherence among women, which was different from previous research results, for the child-rearing/cooking/housekeeping burden. For this statement to be true, the housekeeping burden should be specific to the country or culture that of the subject population. Please support evidences on this interpretation that gender roles have affected the results to be different from other previous studies.

Thank you. While we understand your comment, this was an assumption, and presented as such. Unfortunately, we haven’t found any studies that directly compare the housekeeping burden of Israeli women as compared to other nations during the pandemic. The research on this topic is emerging.

- Citations are generally missing to statements that require supportive evidences. Please provide appropriate references thoroughly, especially for the introduction section.

Thank you for your comment. The introduction includes 29 references to relevant literature, and almost every other sentence is followed by an appropriate citation. The introduction also includes our hypotheses and explanations of our understanding of the phenomenon and the mechanisms at play, as well as topic sentences at the beginning of most paragraphs, resources for which are included in the following sentences. Unfortunately, without exact pointers to the missing references, we were not able to identify the missing supportive evidence.

Round 2

Reviewer 4 Report

I appreciate the authors' response and their revision of the manuscript.
However, I find some points were not addressed and still requires improvement.

1. I do not see any elaboration on the authors' hypothesis regarding the effect of nationality on the coherence. The authors have responded that it was their hypothesis, and I have suggested to provide elaboration if it were their hypothesis but I do not find any improvement. Without any hypothesis on the nationality effect, the readers cannot be convinced about the reason that the authors perform per-nationality analysis, which can introduce bias to the analysis.
2. The confounding effect is still not sufficiently perceived by the authors but have responded confidently to my suggestion that they have not implied causation. One of the biggest possible confounding effect is related to my comment above, nationality. The authors state that many factors differ based on the nationality: "Jewish and Arabic respondents differed in the characteristics, including age, religiosity, educational level, employment, income, number of cohabitants, and place of residence.". It is concluded that "Jewish", single, and male Israelis are more likely to adhere to the guidelines (line 24-26), but isn't it possible that the adherence be related to "age, religiosity, educational level, employment, income, number of cohabitants, and place of residence" rather than nationality? I do not suggest the authors to perform causal analysis or to reveal "unidentified confounders (line 333)" but do suggest that they interpret and provide their limitation of the current analysis correctly.
3. Please consider elaborating about the univariate/multivariate analysis also on the manuscript.
4. If no previous supporting evidence is present about the gender roles affecting the adherence, current interpretation is not sufficient for the readers to be convinced. Why does the child-rearing, cooking, and housekeeping burden requires a greater reliance on others? How does this specifically inhibit adherence to the guidelines?
5. Citations are still missing for statements that require supportive evidences. Following sentences are a few examples, which there are much more. (Please note the quotation marks)
- Due to the continuously high infection rates, social distancing should be and "has been" implemented as a legislative directive on a global scale
- Since the outbreak of the pandemic, researchers have been "vigorously studying" the differences between people who adhere to social distancing and those who do not, or, in some cases, oppose the distancing.

Minor:
Please do not start the sentence with numbers (line 190)

Author Response

Thank you for investing time and effort into rereading our manuscript and submitting your comments. It is greatly appreciated. 

  1. Thank you for bringing our attention to the lack of clarity regarding the study hypotheses. We made sure to include a better, more detailed explanation in the draft, including regarding the nationalities of the respondents. Please see lines 130-137: "We further hypothesize that higher perceptions of protection and danger will be associated with greater Self-reported Adherence (SRA), and that adherence and SoC will differ between people of different genders, nationalities, occupations, income, education, religion and religiosity and marital status. The hypothesized differences between representatives of the two main nationalities residing in Israel—Jews and Arabs—is based on previous studies that reported poorer health status in the Arab population and differences in health behaviors and well-being among the two groups [33].
  2. Thank you for you comment. We have made sure once again to tone down the language (please see all the highlighted changes in the manuscript), and included an explanation of the limitations of the study and further research needed. The abstract now also carries this note on the limitations, ending with: "Further research is needed to assess the role of these factors in Jewish and Arab populations." As you kindly noted, a path analysis may be beneficial and it is out of our scope. However, our multivariate analysis presented in table 2, does include most the factors you mentioned, namely gender, relationship status, religiosity, education level, family income and age (next to perceptions of protection and danger and SoC). 
  3. Thank you for pointing out to us this need. Linear regression was performed to assess association between SRA and explanatory variables. We added this information to the methods section, line 178-179. 
  4. Thank you for your continuous attention to this aspect of our discussion of the results. We have elaborated further on this assumption: "Among the factors that might have contributed to this finding may be the child-rearing, cooking, and housekeeping burden that falls predominantly on women [42] and requires a greater reliance on others, inhibiting women’s ability to fully adhere to the guidelines. In the Israeli society, child-rearing is to a large extent a communal effort, and relies on the help of family members, including non-cohabiting ones like grandparents [43,44]. Further, if women are more responsible for grocery shopping than men, as is the case in the United States but has not been measured in Israel [45], they may be less likely to fully adhere to the social-distancing guidelines." (lines 287-194).
  5. Thank you for bringing our attention to these deficits. We included proper sources in our manuscript, see lines 38 (cit. 2), and line 43 (cit. 3-6). 

Finally, following your suggestion, we made sure not to begin any sentences with numbers.